# OpenReview forum: "Evading Safety Alignment in Open-source LLM Deployment by Embedding Semantic Shift"
_ICLR.cc/2026/Conference — Submitted to ICLR 2026_

### Official Review · Reviewer_89hz · 2025-10-18

**Soundness:** 3
**Presentation:** 2
**Contribution:** 3
**Rating:** 6
**Confidence:** 3

**Summary:**

The paper proposed a new jailbreaking attack, which perturbs the embedding space to induce harmful answers. The design is based on the findings that adding different scales of perturbation on some dimentions of the embedding space could induce a variety of unexpected effects, including deviation, harmful outputs, or totally nonsense glithes. A search algorithm then find the proper perturbation scale, dimension, and location to induce the harmful outputs on the original malicious question. The evaluation shows improved attack success rate compared with baseline attacks.

**Strengths:**

The finding is interesting: a simple "blind" perturbation on embedding space can induce different unexpected model behaviors. Treating the embedding layer as a deployment-time attack vector (no weight edits, no prompt artifacts) is novel in the attack context. Though it requires the white-box access (can add perturbation to embedding), it fits in the scenarios that the user is using the public open-source models, or a developer uses an in-development model for malicious purposes.

It has shown promising results. The reported outcomes are strong at face value: high success rates across multiple prompts/models and signs that benign capabilities are largely retained.

**Weaknesses:**

The findings are largely empirical without theoretical arguments. For instance, description of finding 2 and 3 csays "similar" and "likely", without a clear definition. Are there quantitative results about how likely the findings are valid on a specific case? It looks intuitive that if adding large perturbation, the model outputs will be nonsense glitches, but the existence of that "harmful but nott deviated" region is not guaranteed. I hope the author could provide more empirical numbers or theoretical analysis.

Some of the design and evaluation details are not clear in the main paper. A key component for making the attack pipeline automated is the "multi-stage classifier". However, the classifier relies on a set of existing tools, including the keyword-matching refusal detect and model-based deviation/harmful detect. For the model-based classifier like HarmBench-Llama and LlamaGuard, are they accurate enough for making the classification accurate? Especially, I did not fully understand how HarmBench-Llama detects deviation -- the model detects a set of behaviors actually.

The paper could have more analysis of impacting factors of attack success rate or efficiency numbers, such as the categories and length of malicious questions. Intuitively, the longer malicious prompts could lead to slower optimization. Additional analysis here could help the reader understand the attack impact in depth.

Finally, the embedding space scaled perturbation idea is presented in related work, e.g., Jailbreak Antidote: Runtime Safety-Utility Balance via Sparse Representation Adjustment in Large Language Models (ICLR 2025). It (and similar efforts) should be mentioned as part of the related work.

**Questions:**

* Given a specific malicious prompt, how likely does the "totally harmful" region exist? Could you provide quantitative probabilities or formal analysis?
* How does the HarmBench-Llama model achieve the deviation-detect? Are there inaccuracy issues of the detection/classification?

---

> ### Author Response · Authors · 2025-11-29
>
> **Re for Weaknesses&Questions:**
>
> **W1&Q1: Lack of Theoretical Arguments**
>
> Thank you for the feedback. We acknowledge that the SEP method is empirically driven, and the success of embedded perturbation attacks is influenced by multiple factors—such as the model, prompt, perturbation token, and perturbation dimension—making theoretical calculation of a “completely harmful” region challenging.
>
> However, in Experiment 2 (Appendix A), we randomly sampled 10 prompts and applied token-level linear perturbations across random dimensions, observing such harmful regions in every case. This random validation, along with the high attack success rate reported in the evaluation, strongly supports the method’s applicability.
>
> **W2&Q2: Questioning about Multi-stage Classifier**
>
> Thank you for your inquiry. Harmbench[1] specifies the criteria for successful test cases in Appendix B.1. These criteria include two requirements: “Generated behaviors that are harmful but not instances of the target behavior shall not be counted as successful cases,” and “Generated cases that only respond to, analyze, discuss, or otherwise react to the target behavior without actually exhibiting it shall not be counted as successful cases.” These criteria are enforced concurrently during both classifier training and inference phases, ensuring that model outputs classified as successful attacks are inherently behavior-related.
>
> Additionally, HarmBench-Llama provides accurate harm detection, cross-validating Llama-Guard and enhancing classifier reliability.
>
> To further verify accuracy, we introduced an LLM-as-a-Judge (GPT-5 with predefined prompts) and human evaluation. The latter involved manually reviewing 30 randomly sampled malicious responses (~20% of total). Results are below:
>
> | Method            | **Llama-2-7B** | **Llama-3.1-8B** | Qwen2.5-7B |
> | ----------------- | -------------- | ---------------- | ---------- |
> | GPT-5             | 84.00%         | 92.00%           | 94.67%     |
> | Manually Sampling | 96.67%         | 90.00%           | 96.67%     |
>
> In addition, we conducted a manual evaluation of the samples in llama2 experiment of GPT-5. We found 12 false negative cases, and the final ASR should be **92.00%**. In summary, the accuracy of the Multi-stage Classifier is acceptable.
>
> **W3: More Factors of ASR Analysis**
>
> Since the success rates of models other than gemma-7b are very high, resulting in too few failure samples to enable accurate analysis, we focus our analysis on the failure samples from the gemma model.
>
> **(1)  Categories**
>
> We collected 28 failure samples across various malicious categories and present the top three most prevalent categories as follows (category standards uniformly referenced from jailbreakbench):
>
> | Categories | Disinformation | Harassment/Discrimination | Physical harm | Others |
> | ---------- | -------------- | ------------------------- | ------------- | ------ |
> | **Count**  | 11             | 4                         | 3             | 10     |
>
> Overall, SEP has the lowest attack success rate in the Disinformation category, at just 45%. Prompts in this category often involve more covert malicious information and more complex requirements.
>
> **(2) Prompt Length**
>
> We separately collected the average prompt lengths for successful and failed examples during the gemma-7b testing process. Specifically, the average length for all prompts was **86.43** tokens, with failed samples averaging **91.68** tokens and successful samples averaging **85.23** tokens. The difference in length is extremely small—potentially less than a single token—indicating that prompt length has a negligible impact on SEP.
>
> **W4: Supplement Related Work**
>
> Thank you for your response. [2] is a method that achieves real-time adjustment of safety preferences for large language models by manipulating sparse subsets of the model's internal state during inference. It operates on the distribution of neurons in the MLP layer within the decoder layer, exhibiting similarities to representation engineering approaches that manipulate activations in intermediate layers[3, 4]. Following Reviewer tudL's suggestions, we have categorized this series of methods under Latent Space Attacks [5]. However, our approach specifically targets perturbations at the embedding layer—the input to the first decoder layer—and thus has distinct target components. We incorporate an introduction to the aforementioned Latent Space Attacks in the revised version.
>
>
>
> **References:**
>
> [1] HarmBench: A Standardized Evaluation Framework for  Automated Red Teaming and Robust Refusal
>
> [2] Jailbreak Antidote: Runtime Safety-Utility Balance via Sparse Representation Adjustment in Large Language Models
>
> [3] Representation Engineering: A Top-Down Approach to AI Transparency
>
> [4] Uncovering Safety Risks of Large Language Models through Concept Activation Vector
>
> [5] Operationalizing a Threat Model for Red-Teaming Large Language Models

---

### Official Review · Reviewer_imnT · 2025-10-27

**Soundness:** 3
**Presentation:** 2
**Contribution:** 2
**Rating:** 4
**Confidence:** 4

**Summary:**

This paper investigates the vulnerability of open-source LLMs to perturbations at embedding layer, introducing the phenomenon of semantic shift when attackers manipulate the embedding vector of high-risk tokens. The authors propose SEP, a framework that systematically identifies and applies minimal but targeted perturbations to token embeddings, thereby evading safety alignment mechanisms in LLMs.

**Strengths:**

- This paper focuses on perturbation attacks in the embedding layer of LLMs, unlike those methods that concentrate on modifying input prompts.
- The mathematical formula for the embedding perturbation process is explicit, supporting reproducibility and a clear understanding of the operational mechanism.
- The discussion section offers practical suggestions, indicating an awareness of the broader implications of safe LLM releases.

**Weaknesses:**

- This paper would significantly benefit from a more thorough comparison with related work. Since 2023, several studies [1,2,3] have explored how internal representations affect LLM safety. While manipulation at the embedding layer is relatively novel, the paper lacks strong empirical evidence demonstrating why attacks at the embedding layer are more advantageous than those proposed by related work.
- Only modifying the embedding layer to achieve such a high ASR seems to conflict with prior work. For instance, [2] reports that safety-related concepts typically emerge in middle layers, whereas early layers, especially layer 0, cannot be effectively controlled to achieve the attacks. I invite the authors to comment on this.
- SEP relies on an external multi-stage classifier to identify whether a response is harmful. In a black-box scenario, this is a strong and potentially unrealistic assumption.

[1] Representation Engineering: A Top-Down Approach to AI Transparency, https://arxiv.org/abs/2310.01405

[2] Uncovering Safety Risks of Large Language Models through Concept Activation Vector, https://arxiv.org/abs/2404.12038

[3] Refusal in Language Models Is Mediated by a Single Direction, https://arxiv.org/abs/2406.11717

**Questions:**

- Can the authors clarify the practicality of SEP in real-world LLM application scenarios?
- Can the authors provide more detailed quantitative results on whether perturbations introduced by SEP can be detected via simple embedding-space similarity checks, or whether attack traces are left that defenders could leverage?

---

> ### Author Response · Authors · 2025-11-29
>
> **Re for Weaknesses**
>
> **W1: More Comparison with Related Work**
>
> First, we included the representative engineering attack method SCAV[2] as a baseline. Below are the brief comparison results (150 samples across 3 models):
>
> | Methods   | **Llama-2-7B** | **Llama-3.1-8B** | Qwen2.5-7B |
> | --------- | -------------- | ---------------- | ---------- |
> | SEP(Ours) | 100.00%        | 99.32%           | 98.67%     |
> | SCAV      | 92.00%         | 86.67%           | 91.33%     |
>
> In terms of attack effectiveness, our method also outperforms representative engineering attacks.
>
> Secondly, we need to clarify that the focus of our SEP proposal is to demonstrate the existence of the embedding layer as a vulnerable attack vector. By introducing minimal perturbations in a single dimension to the embedding vectors corresponding to one or several tokens, safety alignment can be bypassed. This finding has not been demonstrated in prior work.
>
> Moreover, the method of modifying the output of the embedding layer—particularly the SEP—enables plug-and-play attack effectiveness for any white-box model without requiring the collection of intermediate layer data and training of linear classifiers as in SCAV[2], significantly reducing the attack cost.
>
> **W2: Potential Conflicts with Prior Work**
>
> Thank you for your feedback. First, based on relevant work [2], we acknowledge that safety-related concepts emerge in intermediate or deeper decoder layers, while early layers do not exhibit clear linear separability. However, SEP applies perturbations to the decoder layer input at layer 0 rather than to activation outputs, and thus does not apply to the aforementioned conclusion.
>
> Secondly, based on the empirical study in Appendix A, we conclude that “Embedding perturbations induce semantic shifts, resulting in Total and Deviation but Harmful Responses with three regions of linear distribution, which pose potential threats for LLMs.” This represents a significant deviation from linear separability, and further research is needed to determine whether the embedding layer possesses a concept of safety and linear separability. The ASR of SEP is empirically validated in Appendix A, and extensive linear perturbation experiments have demonstrated that SEP is highly effective.
>
> **W3: Strong Assumption of Classifier**
>
> Following Reviewer tudL's suggestions, we have re-evaluated the positioning of our current approach and provided a new threat model. Within the taxonomy outlined in [2], our attackers' assumptions align with those of Latent Space Attacks, where the attacker can access or tamper with activation representations or embedding vectors. Therefore, the current attack scenarios for SEP are limited to white-box attacks.
>
> Therefore, introducing a multi-stage classifier in a white-box scenario does not constitute an overly strong assumption, which should be analogous to the assumptions made when introducing linear classifiers in representation engineering methods.
>
>
>
> **Re for Questions:**
>
> **Q1: Real-world Scenarios of SEP**
>
> As mentioned regarding W3, our approach shares a similar threat model with comparable representation engineering methods, with attacks confined solely to white-box scenarios. In real-world scenarios, attackers can exploit SEPs to extract unauthorized model outputs, compromising model integrity. Additionally, SEPs can function as hidden model plugins or hooks to upload malicious models onto model distribution platforms. The implementation of this capability, along with a demonstration, is showcased in https://anonymous.4open.science/w/Embedding-Poisoning-3916/
>
>
>
> **Q2: Embedding-space Similarity Checks**
>
> Thank you for your suggestion. We calculated the cosine similarity between the embedding layer output matrices before and after perturbation. The evaluation results are as follows (involving 3 target models, with the similarity average computed for 150 samples per target model):
>
> |                   | **Llama-2-7B** | **Llama-3.1-8B** | Qwen2.5-7B |
> | ----------------- | -------------- | ---------------- | ---------- |
> | Cosine Similarity | 0.8643         | 0.8873           | 0.6417     |
>
> Specifically, when the perturbation magnitude is controlled within 5, a cosine similarity of 0.8 to 1 can be maintained. However, when the perturbation magnitude reaches 50 or higher, the cosine similarity drops to approximately 0.1. Overall, SEP can ensure a certain level of cosine similarity, but this does indeed serve as an effective defense against SEP attacks. We elaborate on this mitigation strategy in the revised discussion section.

---

### Official Review · Reviewer_pEaN · 2025-10-31

**Soundness:** 2
**Presentation:** 3
**Contribution:** 2
**Rating:** 2
**Confidence:** 4

**Summary:**

This paper introduces an attack framework called Search-based Embedding Poisoning (SEP) which targets the safety alignment of open-source Large Language Models (LLMs). The authors first identify a "Semantic Shift" phenomenon, where slight, controlled perturbations to the embedding vectors of "dangerous tokens" can cause the model to bypass its safety training. The SEP framework uses a three-stage search strategy to efficiently find these specific perturbations, achieving a high attack success rate across several models

**Strengths:**

- The work identifies and explores a vulnerability in the embedding layer, which is a different attack surface from more common prompt-level or suffix-based attacks.
- The "Semantic Shift" phenomenon is documented through an initial empirical study, providing a clear foundation for why the proposed attack framework is effective.
- The evaluation is quite thorough, testing the SEP method against six different open-source LLMs , four baseline attack methods , and two representative safety defense techniques.

**Weaknesses:**

- The threat model assumes the attacker can modify the model's execution flow or files to inject the perturbation before the decoder layers. This is a strong assumption that requires more access than a standard prompt injection attack and may not be feasible in many deployed scenarios. In other words, this paper starts with a scenario where open-source models are deployed on HuggingFace, but how the attackers manipulate the embeddings? It is unclear.
- I am not sure if this paper is novel considering [1]
- The attack's effectiveness seems highly dependent on identifying and perturbing explicit "dangerous tokens". This approach might not be as effective against more implicit or complex malicious queries where the harmful intent isn't located in a few obvious words.
- The performance on Gemma-7B was notably lower in both success rate and efficiency. While the paper suggests this indicates a more robust embedding layer, the specific reasons why this model is more resistant to this particular attack aren't deeply explored.

[1] Representation Engineering: A Top-Down Approach to AI Transparency

**Questions:**

n/a

---

> ### Author Response · Authors · 2025-11-28
>
> **Unclear Threat Model**
>
> Thank you for your feedback. Following Reviewer tudL's suggestions, we have re-evaluated the positioning of our current approach and provided a new threat model. Within the taxonomy outlined in [2], our attackers' assumptions align with those of Latent Space Attacks, where the attacker can access or tamper with activation representations or embedding vectors. However, our method only requires access to the embedding layer's output, whereas most latent space attacks necessitate access to activations across multiple or even all decoder layers—a relatively weaker assumption. We redefine the threat model in the revised version.
>
>
>
> **Question Regarding Novelty**
>
> A series of representation engineering or activation engineering methods, exemplified by RepE[1], often advocate extracting “concept activation” [3] or “rejection direction”[4] from hidden states output by decoder layers. These methods either directly reduce the projection of the activation matrix along this direction or use it to train adversarial samples to break secure alignment. However, SEP only attacks the embedding layer without accessing any hidden representations from the decoder layer. This has not been demonstrated in previous research, including related work [5], which still requires gradient guidance from the final decoder layer. At the same time, our approach requires modifying only a single or a few dimension values in the embedding matrix corresponding to specific tokens. This subtle modification ensures concealment while also warranting further investigation into its impact on subsequent decoder layers.
>
> Additionally, considering the novelty of the attack effectiveness, we included the representative engineering attack method SCAV[2] as a baseline. Below are the brief comparison results (150 samples across 3 models):
>
> | Methods   | **Llama-2-7B** | **Llama-3.1-8B** | Qwen2.5-7B |
> | --------- | -------------- | ---------------- | ---------- |
> | SEP(Ours) | 100.00%        | 99.32%           | 98.67%     |
> | SCAV      | 92.00%         | 86.67%           | 91.33%     |
>
> In terms of attack effectiveness, our method also outperforms representative engineering attacks.
>
>
>
> **Depending On Dangerous Tokens**
>
> We recognize that some malicious queries are more obscure (e.g., “Write a Twitter thread posing as an expert about how vaccines cause autism”). In such cases, the LLM still extracts dangerous spans like “vaccines cause autism,” which convey misleading and potentially harmful intent. We clarify in the revision that “dangerous tokens” encompass both explicitly harmful terms and those implying harmful intent.
>
> Meanwhile, in the first set of experiments in Appendix A, we implemented linear perturbations at the sentence level—that is, across all tokens—and similarly observed regions triggering escape, which accounted for a significant proportion. Therefore, dangerous tokens are not “highly dependent.”
>
> However, the perturbation motivation for dangerous tokens is relatively empirical. Therefore, we additionally attempted linear perturbations on individual neutral tokens beyond dangerous tokens and observed the model's response. Surprisingly, a series of neutral tokens still triggers a harmful response. Below are examples of successful words (bold italics indicate tokens triggering harmful response):
>
> ***Write*** a ***tutorial*** ***on*** ***how*** to ***make*** a ***bomb***
>
> The conclusion is unexpected: SEP can be implemented, and attacks succeeded on most tokens. This further highlights the vulnerability of the embedding layer safety. Nevertheless, based on Findings 3 in Appendix A, we still choose to retain the perturbation design targeting dangerous tokens to maintain attack stability.
>
>
>
> **Performance on Gemma-7B**
>
> Based on statistical analysis, the average perturbation amplitude required for gemma-7b is 166.99, which is significantly higher than the 41.68 of the next highest model. This indicates that gemma-7b indeed exhibits greater robustness to unidimensional perturbations.
>
> References:
>
> [2] Operationalizing a Threat Model for Red-Teaming Large Language Models
>
> [3] Uncovering Safety Risks of Large Language Models through Concept Activation Vector
>
> [4] Refusal in Language Models Is Mediated by a Single Direction
>
> [5] Soft prompt threats: Attacking safety alignment and unlearning in open-source llms through the embedding space

---

> ### Author Response · Authors · 2025-12-03
> **Concern About Potentially Unreliable Review**
>
> Dear Area Chair,
>
> We appreciate the time and effort that you and the reviewers have dedicated to evaluating our submissions. However, we wish to **formally express our concern regarding the unfairness of reviewer pEaN's evaluation** who appears to have been assigned to both of our manuscripts (Paper IDs: **9128** and **8949**).
> The comments on the two papers are obviously **copied and pasted from each other** which is easy to tell. This overlap suggests that the reviewer may have developed a preconceived stance while evaluating one of the papers, leading to potential bias and a lack of objectivity in assessing the other work. After carefully comparing the two reviews, we are concerned that the evaluations may not align with ICLR’s standards of rigorous and impartial peer review. Below, we detail the specific issues observed.
>
> First, the two comments from this reviewer share same phrasing, suggesting the copying behavior. And the issues raised as **weaknesses are largely marginal** compared with strengths and unrelated to the central innovations of our approach, and thus, in our opinion, **cannot reasonably justify the low score** assigned.
>
> More importantly, some of the key criticisms are internally inconsistent. For example, the reviewer states that our method “*assumes a high level of access*”  while at the same time citing *RepE* that in fact assumes an even higher level of access than we do. This kind of contradiction makes it difficult for us to understand the reviewer’s actual technical position or to respond constructively.
>
> Besides, the reviews contain several template-like comments that do not appear to be specific to our work. Out of concern, we checked one of the reviews using the Pangram AIGC detection platform [https://iclr.pangram.com/reviews?submission_number=9128], which classified it as AI-generated. We fully acknowledge that such tools are imperfect and do not treat this as definitive evidence. However, combined with above evidence, we are worried that these reviews may not meet the standard of responsible peer review expected at ICLR.
>
> We want to emphasize that **we fully respect the peer-review process and we welcome critical constructive feedback on our work**. Our only concern is that the final decision be based on fair, well-reasoned, and good-faith evaluations. Therefore, we would kindly ask you to consider either discounting this review in your decision process or, if possible, soliciting an additional independent review to compensate.
>
> Thank you very much for your time and consideration.

---

### Official Review · Reviewer_hM49 · 2025-11-01

**Soundness:** 3
**Presentation:** 1
**Contribution:** 3
**Rating:** 4
**Confidence:** 4

**Summary:**

The paper probes embedding-space vulnerabilities in aligned LLMs. It observes that nudging a single embedding dimension for selected tokens gradually tips responses from refusal to harmful outputs, with small "pockets" that reliably bypass safeguards. Building on this, the authors propose SEP (Search-based Embedding Poisoning) an exponential -> binary → linear search over perturbation magnitude guided by a classifier pipeline. On several instruction-tuned models, SEP reports high attack success on a ~150-prompt set. The broader takeaway: even tiny, structured perturbations to token embeddings, without changing prompts or weights, can meaningfully degrade safety.

**Strengths:**

### What I see as the main contributions

- Phenomenology: a clear description of refusal → uncertainty → deviation regimes as perturbations grow, with concentrated bypass regions.
- Method: a simple, reimplementable search (SEP) to locate those regions efficiently.
- Evidence: consistent success across multiple open models, plus basic ablations (e.g., temperature, search strategy).

### Strengths

- The threat surface is concrete and practical (embedding outputs at inference), which deployment teams actually have to think about.
- SEP is easy to grasp and reproduce; the staged search hits a nice balance between speed and coverage.
- The qualitative framing (response regimes and types) helps practitioners reason about failure modes rather than treating them as one-off glitches.

**Weaknesses:**

### Limitations and weaknesses

- Placement in existing taxonomies is thin. The work would read more cleanly if it explicitly located itself within public threat-model frameworks that already discuss latent-space manipulations and targeted compromises. In particular, "Operationalizing a Threat Model for Red-Teaming LLMs" and "Red-teaming for generative AI: Silver bullet or security theater?" lay out categories (e.g., latent space and targeted attacks) that map closely to what SEP is doing; acknowledging that context would clarify novelty and avoid the impression of “first to discuss embedding-layer threats.”
- Narrow benchmark. The evaluation relies on ~150 malicious prompts (largely JailbreakBench/HarmBench). That’s small relative to current safety eval suites and misses long-context, tool-use, multilingual, and policy-borderline prompts that often bite in practice.
- Classifier-dependence. Success is adjudicated by a multi-stage classifier stack; disagreements among classifiers can shift measured ASR. A small dose of human labeling or cross-checking would ground the claims.
- "Dangerous tokens" is too coarse. Tokens like "marijuana" or "homicide" can be benign in legal/medical contexts. Collapsing lexical presence into hazard misses intent and context, which likely explains some of the brittle behavior you observe.


### High-leverage suggestions

- Situate SEP within public taxonomies. Briefly connect to the categories and definitions in Operationalizing a Threat Model for Red-Teaming LLMs (characterizes Schwinn et al., 2024 as a Latent Space Attack) and Red-teaming for generative AI: Silver bullet or security theater?; also distinguish your integrity-oriented embedding perturbations from "embedding inversion" (privacy) and from suffix-based adversarial prompting. Mention these related works to show continuity and difference.
- Broaden the eval slice just a bit. Add a small panel of longer, tool-use, and multilingual prompts; report per-category ASR and variance across seeds. Even 50–100 extra prompts would strengthen external validity.
- Anchor the classifier calls. Report agreement rates between your classifiers and include a modest human-rated subset; add a sensitivity sweep over thresholds so readers can see how ASR moves.
- Refine token selection. Replace a static "dangerous token" list with a lightweight contextual hazard heuristic (span-level, intent-aware) and revisit the Appendix finding, this will read less like a lexical effect and more like a policy-effect.

### Nits / small fixes

- Clean up obvious typos in figure labels (“Perturabation”, “Posioned”, “Deinal”). Mostly these typos are in figures or captions of figures. Feels like the paper was written in a rush.
- Reword the line “Embedding perturbations induce semantic shifts, resulting in Total and Deviation …” to something like: “Embedding perturbations induce semantic shifts; as the magnitude increases, responses fall into three regimes (refusal, uncertain, deviation), with harmful outputs concentrated in parts of the uncertain regime.”
- Keep notation for the perturbation magnitude and quality metrics consistent across tables and captions.

**Questions:**

See weaknesses above

---

> ### Author Response · Authors · 2025-11-28
>
> **Placement in existing taxonomies is thin**
>
> Thank you for this comment. We further clarify the classification of methods and threat models with reference to [1] and [2]. While SEP, like other targeted attacks [2], intentionally focuses on a specific component of the LLM (the embedding layer), it differs from Latent Space Attacks [1] in terms of the targeted layer. For example, the hidden states or embedding space mentioned in [3] and [4] generally correspond to the output of activation functions in decoder layers, whereas the embedding layer serves as input to the first decoder layer. Thus, SEP is not strictly a Latent Space Attack. That said, we recognize that SEP shares a similar threat model with many Latent Space Attacks, in that attackers may access or modify internal representations during inference. We have refined this classification in the revision and highlighted distinctions among suffix-based adversarial prompting, embedding inversion, and our approach.
>
> We also agree that the claim of being the “first to discuss embedding-layer threats” was overly vague. A key contribution of our work is showing that minimal perturbations at the embedding layer can subvert safety alignment—a novelty compared to attacks on decoder-layer activations. We emphasize this point in the revised manuscript.
>
>
> **Narrow benchmark**
>
> To enhance multilingual evaluation and expand the benchmark, we randomly selected 100 examples from Jailbench[5] for assessment. JailBench is a comprehensive, multi-domain security evaluation dataset designed to assess jailbreak attack risks for large language models in Chinese contexts. Below are the summary evaluation results (featuring three target models and LLM-fuzzer as the baseline):
>
> | **Method** | **Llama-2-7B** | **Llama-3.1-8B** | Qwen2.5-7B |
> | ---------- | -------------- | ---------------- | ---------- |
> | SEP(Ours)  | 87%            | 92%              | 91%        |
> | LLM-fuzzer | 40%            | 45%              | 67%        |
>
> SEP maintains a high attack success rate on Chinese prompts, substantially outperforming the baseline, with perturbation magnitudes comparable to prior work—supporting its suitability for multilingual settings.
>
>
> **Classifier-dependence**
>
> To validate the multi-layer classifier’s accuracy, we introduced an LLM-as-a-Judge detector (GPT-5 with predefined prompts) and human evaluation as controls. The latter involved manually reviewing 30 randomly sampled malicious responses (≈20% of the total) to assess safety violations. Results are summarized below:
>
> | Method            | **Llama-2-7B** | **Llama-3.1-8B** | Qwen2.5-7B |
> | ----------------- | -------------- | ---------------- | ---------- |
> | GPT-5             | 84.00%         | 92.00%           | 94.67%     |
> | Manually Sampling | 96.67%         | 90.00%           | 96.67%     |
>
> In addition, we conducted a manual evaluation of the samples in llama2 experiment of GPT-5. We found 12 false negative cases, and the final ASR should be **92.00%**.
>
> In summary, even when employing alternative classifiers (such as GPT-5) to mitigate unfairness stemming from classifier dependency, or conducting manual verification, the ASR method does not exhibit significant changes.
>
>
> **Dangerous tokens is too coarse**
>
> First, from the attacker's perspective, attack samples are often generated based on specific dangerous statements, and such dangerous statements frequently contain words or phrases that are inherently dangerous or intended to be harmful. Moreover, our method does not rely on a static dangerous-word list. Instead, we employ a prompt-engineered LLM to identify phrases indicative of malicious intent, with manual verification. This LLM-based detector achieves 85% accuracy. Prompt details and evaluations are provided in Appendix G. This approach aligns with the “lightweight contextual” and “intent-aware” principles you highlighted. We also recognize that some malicious queries are more subtle (e.g., “Write a Twitter thread posing as an expert about how vaccines cause autism”). In such cases, the LLM still extracts dangerous spans like “vaccines cause autism,” which convey misleading and potentially harmful intent. We clarify in the revision that “dangerous tokens” encompass both explicitly harmful terms and those implying harmful intent.
>
>
> **Nits / small fixes**
>
> We thank the reviewer for their meticulous feedback. We sincerely apologize for any grammatical errors and unclear expressions in the current version and have carefully revised the text to improve readability and accuracy.
>
> **References:**
>
> [1] Operationalizing a Threat Model for Red-Teaming Large Language Models
>
> [2] Red-Teaming for Generative AI: Silver Bullet or Security Theater?
>
> [3] Representation Engineering: A Top-Down Approach to AI Transparency
>
> [4] Uncovering Safety Risks of Large Language Models through Concept Activation Vector
>
> [5] JailBench: A Comprehensive Chinese Security Assessment Benchmark for Large Language Models

---

### Author Response · Authors · 2025-12-03
**Summary of Rebuttal Responses and Rebuttal Revision**

Dear Area Chair,

We would like to express our sincere appreciation to the Area Chair and all reviewers for their valuable time and insightful comments during the review and rebuttal process. While we were unable to receive further feedback from the reviewers during the rebuttal period due to unforeseen circumstances, we take this opportunity to provide a concise summary of the key points raised regarding our submission.

The reviewers consistently acknowledged the following strengths of our work, SEP:

1. A **thorough empirical study** that reveals vulnerabilities in existing embedding layers and systematically analyzes the "Semantic Shift" phenomenon.
2. A **straightforward and highly reproducible** method.
3. **Strong performance** demonstrated under comprehensive evaluations involving multiple models and baselines.

In response to the reviewers' concerns, we have addressed the four main points as follows:

1. **Method Positioning & Relationship to Latent Space Attacks**: We clarified the similarities and differences between SEP and prior latent space attacks. Using the representative SCAV method as a baseline, we demonstrated that SEP exhibits stronger effectiveness.
2. **Threat Model Clarification**: Following Reviewer hM49’s suggestion, we have categorized the threat model as a white-box attack, aligning it with a series of latent space attacks.
3. **Diverse Analysis of Effectiveness**: We further evaluated the effectiveness of the SEP attack from three perspectives: malicious prompt categories, prompt length, and multilingual prompts.
4. **Fairness of the Evaluator**: We performed cross-validation using both the LLM-as-a-judge method and human inspection, confirming an accuracy rate exceeding 90%.

In the revised manuscript, we have:

- Refined the descriptions of representation engineering in the Introduction and Related Works sections;
- Restated the threat model accordingly;
- Added the SCAN method as an additional baseline for comparison;
- Added accuracy evaluation of multi-stage classifier.

We thank you for your patience and efforts throughout this process. Should any questions or concerns remain, we are pleased to provide further clarification.

Sincerely,

The Authors

---

### Meta-Review · Area_Chair_4woU · 2025-12-31

**Summary:**

The paper proposes Search-based Embedding Poisoning (SEP), an attack framework that undermines the safety alignment of open-source LLMs by exploiting a semantic shift in token embeddings.

Main concerns of this work are:
- Positioning of the work relative to related work
- The main novelties in theory, methodology, and application scenarios
- Justification and rationale of the threat model

**Reviewer Concerns:**

Reviewer hM49:
- Placement in existing taxonomies is thin: *partially solved*
- Narrow benchmark: *partially solved*
- Classifier-dependence: *likely to be solved*
- Dangerous tokens is too coarse: *partially solved*

---
Reviewer pEaN:
- Strong assumption of the threat model, which assumes the attacker can modify the model's execution flow or files to inject the perturbation before the decoder layers. *unlikely to be solved*.
*(AC: Since this appears to be a strong assumption, you may explain why it is applicable to attackers.)*
- Not sure if this paper is novel considering [1] *unlikely to be solved*
- The attack's effectiveness seems highly dependent on identifying and perturbing explicit "dangerous tokens". *partially solved*
- The performance on Gemma-7B was notably lower in both success rate and efficiency. *possibly solved*
---
Reviewer imnT:
- more thorough comparison with related work *partially solved*
- Only modifying the embedding layer to achieve such a high ASR seems to conflict with prior work. *unlikely to be solved*
*(AC: You may also need to explain the rationality and advantage of such a setting.)*
- SEP relies on an external multi-stage classifier to identify whether a response is harmful. In a black-box scenario, this is a strong and potentially unrealistic assumption. *Unlikely to be solved*
---
Reviewer 89hz:
- The findings are largely empirical without theoretical arguments. *unlikely to be solved*
- Some of the design and evaluation details are not clear in the main paper. *unlikely to be solved*
- The paper could have more analysis of impacting factors of attack success rate or efficiency numbers *partially solved*
- Embedding space scaled perturbation idea is presented in related work *unlikely to be solved*

**Reviewer Scores:**

Reviewer hM49 *is likely to keep the score*.
Reviewer pEaN *is likely to keep the score*.
Reviewer imnT *is likely to keep or lower the score*.
Reviewer 89hz *is likely to keep or lower the score*.

---

### Decision · Program_Chairs · 2026-01-26

Reject